# A Study on the Improvement of Using Raw Lacquer and Electrospinning on Properties of PVP Nanofilms

**DOI:** 10.3390/nano10091723

**Published:** 2020-08-31

**Authors:** Kunlin Wu, Ding Zhang, Minghua Liu, Qi Lin, Bing-Chiuan Shiu

**Affiliations:** 1Fujian Engineering and Research Center of New Chinese Lacquer Materials, Ocean College, Minjiang University, Fuzhou 350108, China; kunlinwu2020@163.com (K.W.); dingzhang1874@126.com (D.Z.); 2College of Environment and Resources, Fuzhou University, Fuzhou 350108, China; mhliu2000@fzu.edu.cn; 3College of Materials Science and Engineering, Fuzhou University, Fuzhou 350108, China

**Keywords:** raw lacquer (RL), polyvinyl pyrrolidone (PVP), electrospinning, nanofibers

## Abstract

Raw lacquer (RL), ethanol being used as the solvent, was added to polyvinyl pyrrolidone (PVP) and then electrospun into RL/PVP nanofilms. Manufacturing parameters such as RL/PVP ratio, voltage, flow velocity, needle type, and the distance between syringe and the collection board were systematically investigated. A scanning electronic microscope (SEM) was used to observe the surface morphology of nanofilms; the block drop method was used to measure the water contact angle; the mechanical properties of RL/PVP nanofilms of different proportions were tested by universal material testing machine; and Fourier-transform infrared spectroscopy (FT-IR) was used to characterize the structure. Based on the water resistance and acid resistance measurements, the proposed nanofilms demonstrated to be water and acid resistant were successfully produced. The results show that PVP that melts in water becomes incompatible with water after adding raw lacquer, and the acid resistance is greatly improved. Furthermore, the smaller the fiber diameter, the better the mechanical properties of the nanofilms are under low ratio of RL/PVP. With a high proportion of RL/PVP, the inner structure of the nanofilm is denser, and the water resistance and acid resistance are better. The dense structure can protect the inner material of the nanofilms.

## 1. Introduction

Raw lacquer (RL)—commonly called as Chinese lacquer—is a natural polymer matrix composite mainly composed of urushiol (60–65%), water (20–30%), gummy substance (5–7%), glycoproteins (2%) and laccase (0.2%) [1,2]. Raw lacquer is an environment friendly material [3] and is also a primitive coating material in China dated to 8000 years ago. Urushiols—the main components of RL—are catechol derivatives with either a C15 chain or a C17 chain, in which the double bonds may be in a variety of positions. The positions and proportions of unsaturated double bonds may vary within the side chain [4] (see Figure 1). The unique structure of urushiol endues RL many excellent properties, including high luster, high adhesive force to matrices, dense surface that is anti-percolating, stable heat resistance, chemical resistance, an organic solvent, and great antibiosis [5,6,7]. Hence, RL had been used to coat furniture and handcrafts in ancient China and the heritage lacquerware discovered by archaeologist still retain good condition, which suggests that the RL coating possesses durability performance [4]. Nowadays, RL has been involved in numerous fields such as boats, marines, submerged engineering equipment and automobiles [4]. There have been a considerable scholars investigating the chemical structure and properties of RL as well as conducting studies on the related applications in other fields constantly. Gao et al. found that the presence of hexamethylenetetramine (HMTA) in RL had a positive influence on the performances of lacquer films as it improved the morphology, luster, distinctness of image (DOI), hardness, thermal stability and alkali resistance of the coating membranes [8]. Yang et al. incorporated copper (II)-modified montmorillonite (Cu (II)-MMT) with RL, thereby considerably shortening the drying time of resulting membranes in low humidity while improving the gloss and thermal stability of the membranes [9]. Xu et al. used a water-assisted assembly method to produce urushiol-formaldehyde polymer (UFP) membranes that had a porous structure—and thus good hydrophobic properties and acid and alkali resistance [10]. Wang et al. conducted the interaction between urushiol and methylene chloride in order to synthesize urushiol methylene acetal derivatives which produced valuable leading compounds and further improved the design and development of more effective anticancer agents [11]. Kim et al. used urushiol of RL to synthesize a series of poly (methyl methacrylate)-urushiol polymer composition which were then made into membranes with better thermal stability, highly effective antibacterial and antifouling capacity [7].

Electrospinning is an efficient technique that produces polymer nanofibers with a diameter between 5 nm and 500 nm [12]. These nanofibers have high porosity, small pore size, a very large surface-to-volume ratio and good physical and mechanical properties, which make them suitable for biomedical, textile, sensor and energy fields. Therefore, they have great potential in related biomedical science and industry [13,14,15]. In recent years, there have been many studies exploring electrospinning, producing a series of novel nanofibrous materials with application value.

Some researchers have employed electrospinning to develop nanofibrous polymer-based scaffolds. The scaffolds have been used in regenerate human skeletal muscle tissue and in regenerative medicine to transfer growth factor or small molecules, proving the great success of these studies [16,17]. Other researchers have studied the potential application of electrospinning in the food industry. The employment of electrospinning has been substantiated with potential application value in terms of the encapsulation, enzyme immobilization, food coating, filtration materials and active food packaging [18]. Shi et al. expanded and discussed the use of electrospinning in the fields of solar batteries, fuel cells, nano generators, hydrogen energy storage, lithium ion batteries and super capacitors [19,20]. Lee et al. applied melt-electrospinning to produce polypropylene fiber webs and laminates for the protective garments for agricultural workers [21]. Polymer nanofibers have a considerable surface area and volume that strengthens cell adhesion, so antibiotic agent, anticancer agents, proteins, DNA, and RNA can be incorporated with electrospinning stents for drug delivery. Electrospinning also plays a significant role in the tissue engineering field [22,23].

Good results have been achieved in many fields by electrostatic spinning. Polymer nanofibers are made from RL by means of electrostatic spinning, which is a new process of RL film-making and can be applied in more fields. RL appears sticky and demands a long time to dry, hence it cannot be directly used for electrospinning. Diluting RL may result in polymerization, and therefore, RL needs another substance as a carrier. Polyvinylpyrrolidone (PVP) is a nonionic polymer that can be dissolved in absolute ethanol and has good compatibility to serve as a qualified dispersant or an emulsifier. PVP has been commonly used in medicine, cosmetic, food, printing ink, textile, etc. In this study, PVP is used as the carrier that is made into RL-coated nanofibrous membranes through electrospinning, and the resulting membranes are evaluated in terms of water resistance and acid resistance.

## 2. Experimental

### 2.1. Materials

Polyvinylpyrrolidone (PVP; Sigma, St. Louis, MO, US) has an average molecular weight of 360,000. Chinese raw lacquer (RL; Shanxi, China) was filtered with gauze beforehand. NaOH and H_2_SO_4_ (Shanghai Chenghai Chemical Industry Co., Ltd., Shanghai, China) and absolute ethanol (Xilong Science Co., Ltd., Guangdong, China) were all analytically pure.

### 2.2. Preparation of RL/PVP Mixtures

Into a beaker, 0.1 g–0.4 g of raw lacquer (RL) then 10 g of absolute ethanol were infused and mixed for 30 min using a magnetic stirrer. Next, 0.1 g–0.4 g of PVP was slowly added and mixed for 8 h as well in order to prevent polymerization. Afterward, the beaker was sealed with a sealing film throughout the experiment so as to prevent the volatilization of ethanol.

### 2.3. Characterizations

The IR spectra of the membranes were collected by the ATR method using an FT-IR spectrometer (MPIR8400S, Shimadzu, Japan). Thirty-two scans were conducted with a resolution of 4 cm^−1^.

The SEM images of the surface of nanofilms were photographed by a field emission SEM (S3400, HITACHI, Tokyo, Japan) at an acceleration voltage of 5 kV. The SEM images of the cross-section of nanofilms were photographed by a field emission SEM (Nova Nano SEM 230, FEI, Hillsborough, OR, USA) at an acceleration voltage of 2 kV.

Water contact angle (WCA) was measured with 6 μL deionized water at ambient temperature using a dynamic/static contact angle meter (SL200B, KINO, New York, NY, USA). Five different positions were measured on each mesh in order to had the average.

Fiber size distribution: Used Image-Pro Plus 6.0 to mark fiber diameter as much as possible in the image and used Origin to draw graphics.

Mass-loss rate (ω): Dry he nanofilms in vacuum oven at 60 °C until the mass remained unchanged, denoted as m1. Soak the nanofilms in deionized water for 1 h, and then dry it in vacuum oven at 60 °C until the quality remains unchanged, denoted as m2.
(1)$$\omega \;=\;\frac{m1-m2}{m1}\times 100\%$$

The tensile strength (σ): the film was cut into 10-mm-wide (*b*) and 100-mm-long shapes. The universal material testing machine (Instron 1185, Instron, IL, USA) was moved at 5 mm/min to record the peak load (*P*) of the film fracture. The thickness of the membrane was *d*.
(2)$$\sigma \;=\;\frac{P}{bd}$$

### 2.4. Electrospinning

The instrument used for RL/PVP film preparation was a JDF05(Changsha Nayi Instrument Technology Co., Ltd., Changsha, China). RL/PVP mixtures were infused into a syringe. The metallic syringe needle serves as the anode while the collection board was used as the cathode, both of which interact with the externally applied high voltage as Figure 2. The temperature and humidity should be strictly controlled in the experiment. The humidity in the air was ≤60%, and the temperature kept at room temperature. At the feed flow rate of 0.45 mL/h, the horizontal distance between the needle tip and the collecting reel center was 12 cm, and the vertical distance was 10 cm. The needle type was 23G in the international standard, and the corresponding inner diameter was 0.34 mm. Ten grams anhydrous ethanol was used to dissolve the raw paint and PVP of different qualities, and repeated tests were conducted on different samples. The voltage obtained is shown in Table 1. As can be seen from Table 1, when the concentration of the solution increased, the required voltage augmented. A greater concentration may account for the increased viscosity of the solution, meaning a greater force required for the droplet to overcome the surface tension.

## 3. Results and Discussion

### 3.1. Effects of RL/PVP Ratio on Membrane Formation

With 10 g absolute ethanol as the solvent, different RL/PVP ratios were used to form RL/PVP nanofilms. Figure 3 shows the SEM images of RL/PVP nanofilms as related to the RL/PVP ratio. When the mass ratio was 3:1, the fiber structure was a blur because of considerable fiber adhesion. This result may be ascribed to the fact that the ethanol quickly evaporated when the electrospinning nanofibers were ejected and then collected over the collection board. A fiber structure was presented after PVP was dried. However, it took a longer time for the urushiol of RL to dry, and the nanofibers were coated with urushiol so were easily adhered to each other as shown in Figure 3b. With a ratio of 1:1, the resulting films exhibited a clear fiber structure without adhesion. Figure 3d shows that the diameter of fibers was about 200 nm–300 nm. When the ratio was 1:3, the films also show a distinctive fiber structure, but a lower acid resistance, water resistance and mechanical strength, indicating that this ratio failed to reinforce the films.

### 3.2. Effects of Solution Solubility on Film Formation

Based on previous results, RL/PVP of 1/1 provided the films with a distinctive fiber structure. The samples were prepared using this RL/PVP ratio of 1/1 in the following study. The film structure was examined in terms of the solution solubility. A constant amount of ethanol (10 g) was used as a solvent and RL and PVP were added with equal mass (0.2, 0.3 and 0.4 g) that were denoted as RL/PVP = 2/2, 3/3 and 4/4. In Figure 4a, the fiber distribution was concentrated in the range of 0.16–0.22 um, with a small difference in fiber diameters; in Figure 4b, the fiber diameters differ greatly and the fiber diameters were scattered; in Figure 4c, the fiber diameters were relatively large and distributed in a concentrated way. The fiber diameter was increased as a result of an increase in the solution solubility. There are two possible causes. One is that an increase in solution solubility also means that there were more content of RL and PVP over the nanofibers. The other is due to a combination of a rather small voltage and an excessive flow rate, which prevents the nanofibers from being well stretched, and the nanofibers thus demonstrate uneven diameter distribution.

It can be seen from Figure 5 that PVP and RL were at different proportions, and the internal structure of the film was also fibrous. The fiber diameter of Figure 5c,d was larger than those of Figure 5a,b and the internal structure could also be compact, indicating that a higher proportion of RL/PVP had better acid resistance, water resistance and mechanical properties.

### 3.3. Effects of Solution Solubility on Contact Angle

The water contact angle of nanofilms made of different mass ratios of RL to PVP was measured and presented in Figure 6 and Figure 7. When the ratio increased, the change in water contact angle becomes slower. This result was probably ascribed to PVP that was a water-soluble substance. When the nanofilm was in contact with a water droplet, PVP was dissolved immediately whereas urushiol was a hydrophobic material. When the PVP over the surface was all dissolved, the film was then covered by the water repellent material. With the increase of solution RL/PVP content, the denser its internal structure was, the better its water-repellent effect was. Figure 6 shows that the water contact Angle of pure PVP films still remains descending after 30 s while the RL/PVP films exhibit a stabilized water contact angle in 25 s. In addition, Figure 7 shows the difference in water contact Angle between (b) and (d), indicating that the presence of RL improves the performance of PVP nanofilms.

### 3.4. FT-IR Spectra Analysis

Figure 8 shows the FT-IR spectrum of RL, PVP and different RL/PVP films. The absorption peaks of RL are presented as O–H stretching vibration at 3404, 1353, 1278 and 1184 cm^−1^. Afterwards, the addition of PVP did not attenuate the absorption peaks of RL, which suggests that PVP did not interfere with the number of O–H group on aromatic ring of urushiol [8]. By contrast, the carbonyl peak of amide was presented at 1600 cm^−1^, which further substantiated that the presence of RL did not undermine the absorption peak and the basic elements of PVP.

### 3.5. Water Resistance of Nanofilms

Figure 9 shows images of pure PVP and RL/PVP = 4/4 nanofilms that were immersed in deionized water for one hour. The PVP nanofilm was completely dissolved, but RL/PVP nanofilm appears to be water resistant. This result was in conformity with the test results of water contact angle. As can be seen from the mass-loss rate in Table 2, the higher the content of RL/PVP in the solution, the lower the mass-loss rate. This was because the higher the content of RL/PVP in the solution, the denser the structure of RL/PVP nanofilms. Water dissolves the PVP on the surface of nanofilms, but it could not dissolve the PVP inside the nanofilms. This indicates that nanofilms added with RL not only had water resistance, but also can protect the internal substances of nanofilms. Figure 10a is the SEM image after the PVP nanofilm was soaked in water, showing the dense planar structure, which indicates that water immersion only dissolves the PVP over the nanofilm surface, and urushiol was polymerized to form the dense plane. Due to the hydrophobic feature, urushiol provided the interior materials with osmosis prevention. Figure 11 shows that FT-IR spectrum of nanofilm that was immersed in deionized water still exhibits the presence of carbonyl peak of amide at 1600 cm^−1^, which suggests that the RL/PVP nanofilm was not totally dissolved. The composition of nanofilms did not change correspondingly while exhibiting a stable system.

### 3.6. Acid Resistance of Nanofilms

Figure 12 shows the images of RL/PVP = 4/4 nanofilms that are separately immersed in an H_2_SO_4_ solution with different concentrations (i.e., 20% H_2_SO_4_, 40% H_2_SO_4_, 60% H_2_SO_4_, 80% H_2_SO_4_ and 100% H_2_SO_4_) for one hour. The higher the concentration, the darker the shade of nanofilms. Nonetheless, the nanofilms demonstrated excellent acid resistance, which was ascribed to the intrinsic high acid resistance of urushiol. Figure 10 shows the SEM images of RL/PVP = 4/4 nanofilms that demonstrate a dense plane structure after the H_2_SO_4_ acid immersion. During the immersion, PVP could not withstand the acid and was dissolved, which resulted in the agglomeration of urushiol and the formation of a dense plane structure. This structure showed good osmosis prevention to protect the interior of nanofilms. Figure 11 shows the FT-IR spectra of RL/PVP = 4/4 nanofilms where the peak value slightly descends as a result of one-hour immersion in a 100% H_2_SO_4_ solution. This phenomenon indicated that an immersion in a highly concentrated sulfuric acid for a long time adversely affects the structure of urushiol. Conversely, the absorption peaks of RL/PVP = 4/4 nanofilms do not change when immersed in the other solution concentration, which shows that the composition of nanofilms remains intact with a stabilized system. Therefore, the presence of urushiol prevents the nanofilm interior from corrosion as the carbonyl peak of amide was presented at 1600 cm^−1^.

### 3.7. Tensile Strength

It can be seen from Table 3 that the addition of RL had a good effect on the tensile strength of PVP. When RL/PVP ratio was 2/2, the tensile strength of thin films was the best. As the proportion increased, the tensile strength of the film decreased, because the content of RL/PVP in the solution increased, and the diameter of the fibers in the film also increased. The smaller the fiber diameter, the greater the mechanical strength of the fiber. The thicker the fiber, the greater the surface area and the greater the probability of cracking. When external force was applied, the crack in the weakest area expanded rapidly until the whole fiber breaks. Large coarse surface area of fiber diameter greatly increased the possibility of fiber breakage.

## 4. Conclusions

In this study, RL was combined with PVP to being electrospun into nanofibrous films. RL/PVP nanofilms successfully preclude the highly water solubility of pure PVP and are thus highly water and acid resistant. RL/PVP nanofilms would interact with an acid solution and as such to form a dense plane structure that protects the interior. Moreover, adding RL into PVP, the mechanical properties of the film were improved. To sum up, RL/PVP nanofilms can be used as a coating layer that provides waterproof function and acid resistance, thereby increases the service life of the materials that the nanofilms enwrap. This is a pioneering study using raw lacquer and electrospinning technique with an attempt to improve the performance of PVP nanofilms. There are many possibilities for the application of the proposed RL/PVP nanofilms since RL is a natural polymer material that can be further explored in the future. The purpose of this study is to incorporate RL with the electrospinning to protect the interior of composite materials and it is hoped that this study is helpful to the use of raw lacquer in more related fields.

## Figures and Tables

**Figure 1 nanomaterials-10-01723-f001:**
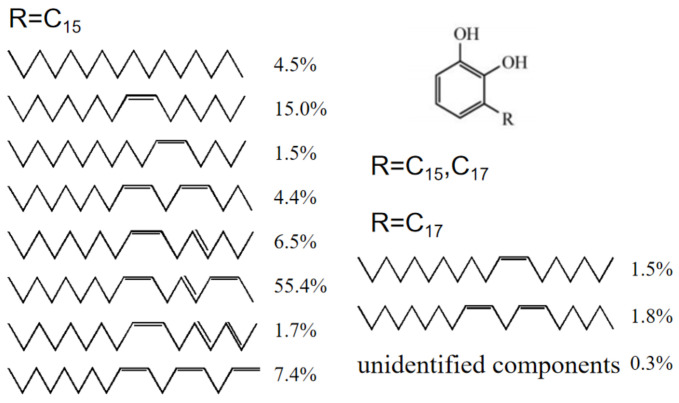
Components and structures of urushiol.

**Figure 2 nanomaterials-10-01723-f002:**
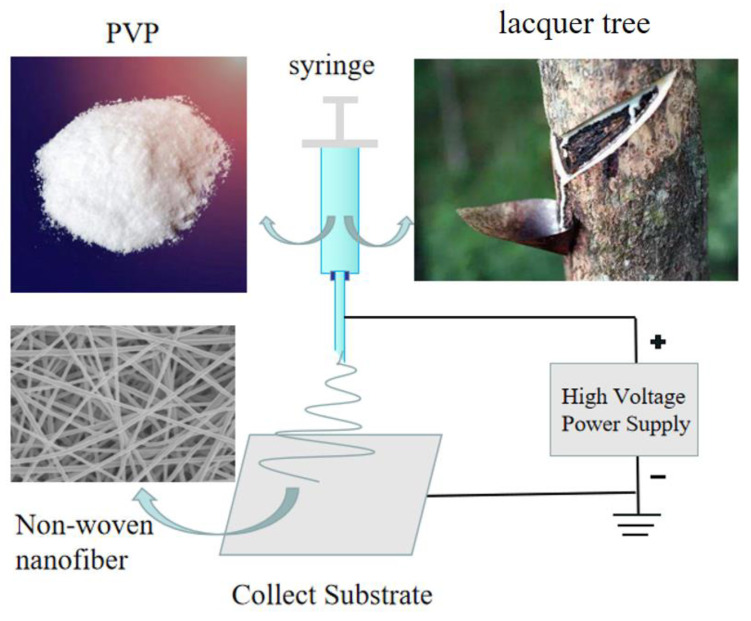
Schematic diagram of electrospinning.

**Figure 3 nanomaterials-10-01723-f003:**
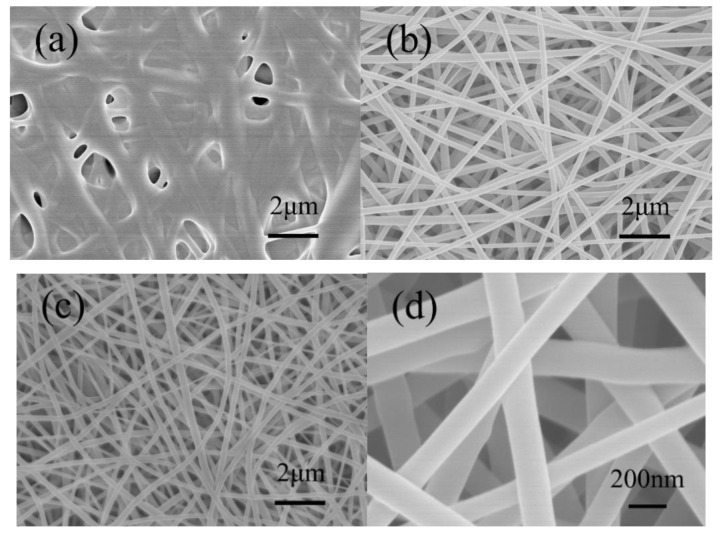
SEM images of nanofilms with different mass ratio of RL/PVP. (**a**) 3:1; (**b**) 2:2; (**c**) 1:3; (**d**) 1:1.

**Figure 4 nanomaterials-10-01723-f004:**
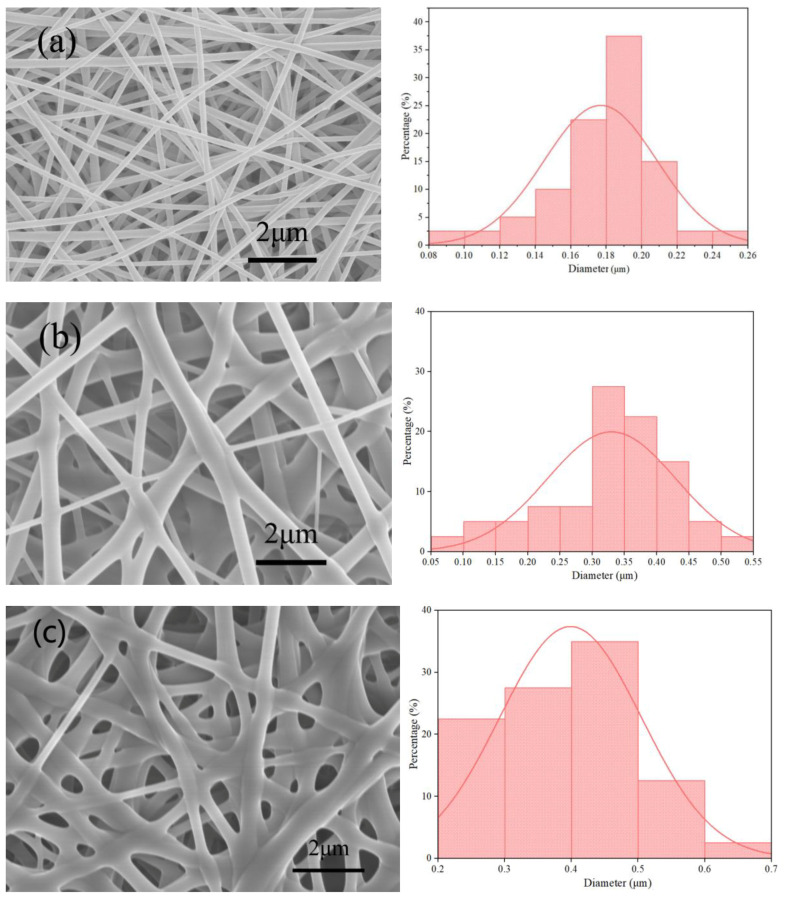
SEM images and diameter of the nanofilms surface as related to mass ratios of RL/PVP. (**a**) 2/2; (**b**) 3/3; (**c**) 4/4.

**Figure 5 nanomaterials-10-01723-f005:**
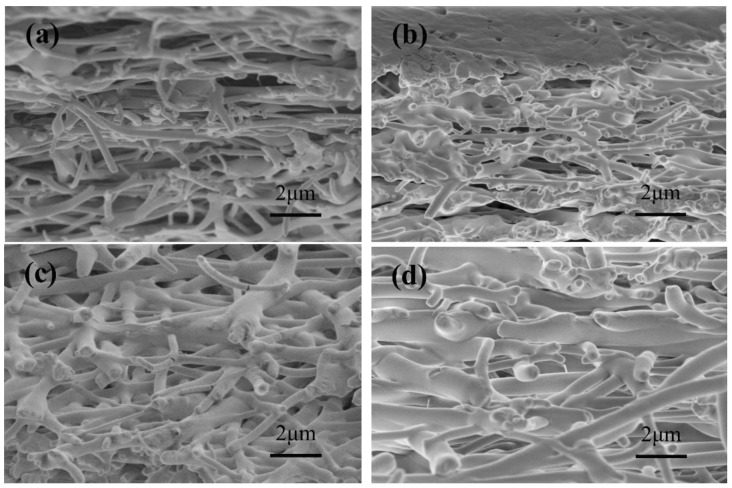
SEM images of the cross-section of nanofilms (**a**) without RL; (**b**) RL/PVP = 2/2; (**c**) RL/PVP = 3/3; (**d**) RL/PVP = 4/4.

**Figure 6 nanomaterials-10-01723-f006:**
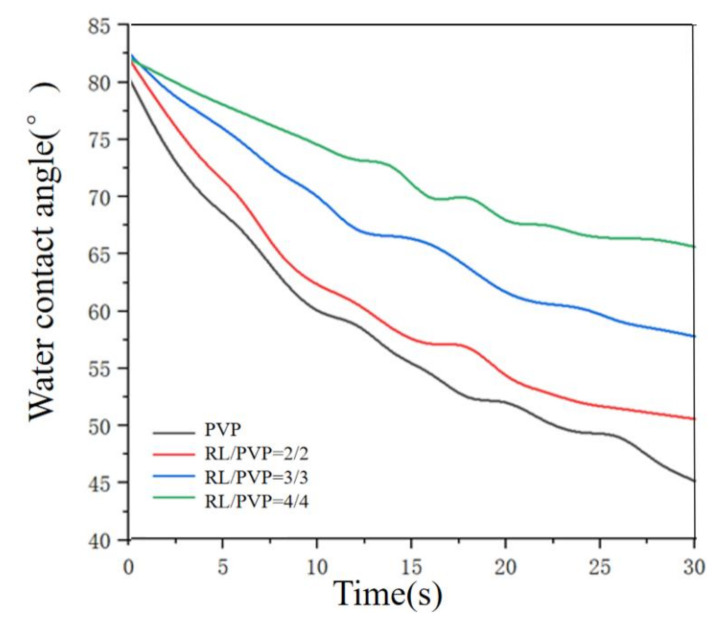
Water contact angle of RL/PVP nanofilms.

**Figure 7 nanomaterials-10-01723-f007:**
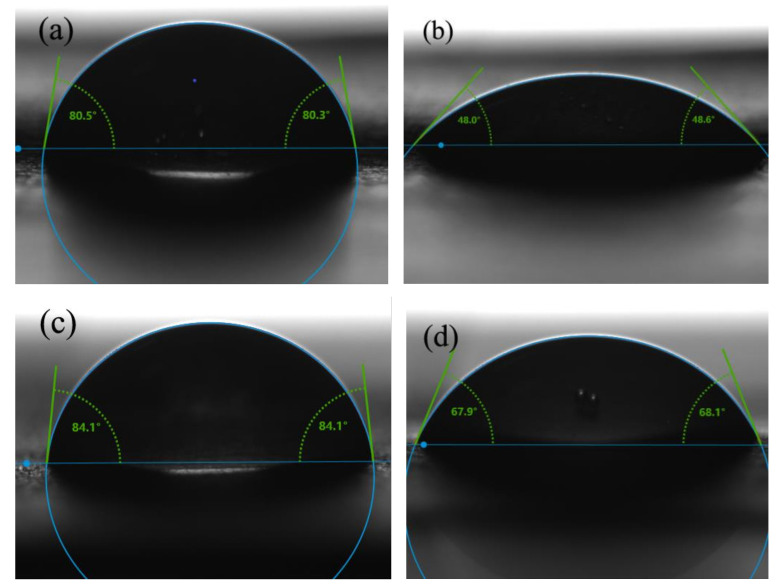
Water contact angle of (**a**) and (**b**) PVP film at 0 s and 30 s, (**c**) and (**d**) RL/PVP = 4/4 film at 0 s and 30 s.

**Figure 8 nanomaterials-10-01723-f008:**
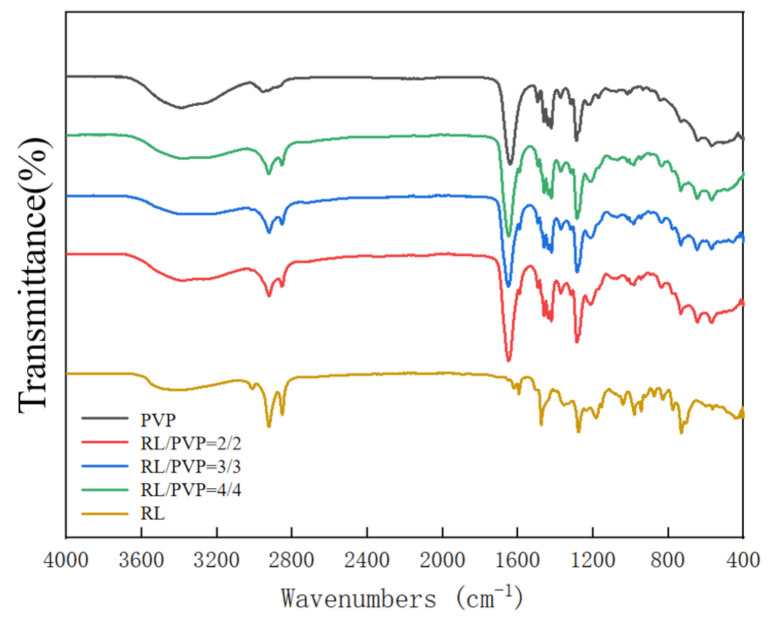
FT-IR spectra of nanofilms as related to the RL/PVP ratio.

**Figure 9 nanomaterials-10-01723-f009:**
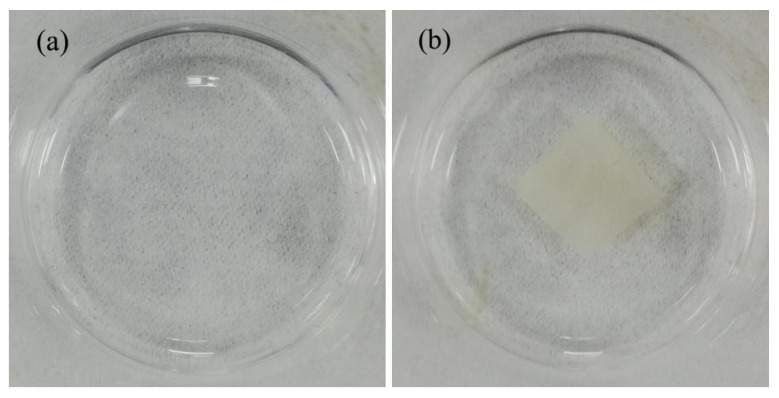
Images of (**a**) pure PVP and (**b**) RL/PVP = 4/4 nanofilms immersed in deionized water.

**Figure 10 nanomaterials-10-01723-f010:**
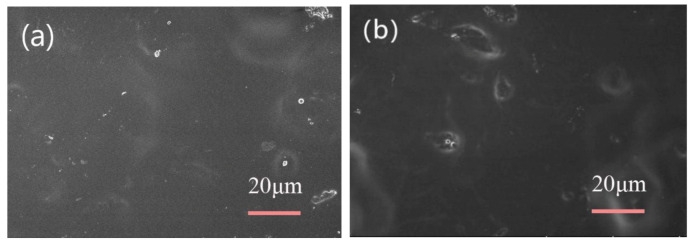
SEM images of nanofilms (RL/PVP = 4/4) after immersion solutions. (**a**) Deionized water; (**b**) 20% H_2_SO_4_; (**c**) 40% H_2_SO_4_; (**d**) 60% H_2_SO_4_; (**e**) 80% H_2_SO_4_; (**f**) 100% H_2_SO. Immersion duration is one hour.

**Figure 11 nanomaterials-10-01723-f011:**
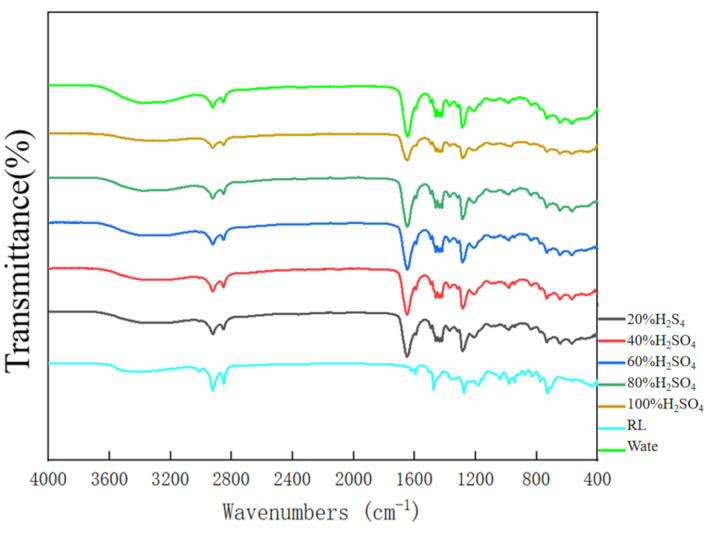
FT-IR spectra of nanofilms (RL/PVP = 4/4) after one-hour H_2_SO_4_ immersion.

**Figure 12 nanomaterials-10-01723-f012:**
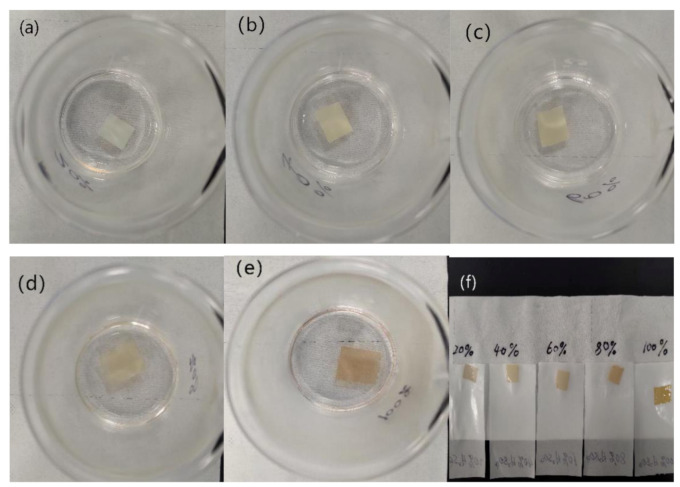
Images of nanofilms (RL/PVP = 4/4) immersed in different acid solutions for 1 h. (**a**) 20% H_2_SO_4_; (**b**) 40% H_2_SO_4_; (**c**) 60% H_2_SO_4_; (**d**) 80% H_2_SO_4_; (**e**) 100% H_2_SO; (**f**) film morphology at different sulfuric acid concentrations.

**Table 1 nanomaterials-10-01723-t001:** Voltage values of different ratios of raw lacquer (RL)/polyvinyl pyrrolidone (PVP).

RL/PVP (g)	Voltage 1 (kV)	Voltage 2 (kV)	Voltage 3 (kV)	The Average(kV)
0/3	15.85	16.46	16.09	16.13
2/2	16.85	17.65	17.06	17.19
3/3	18.32	19.21	18.68	18.73
4/4	21.07	20.46	19.68	20.40

**Table 2 nanomaterials-10-01723-t002:** **** PVP mass-loss rate.

RL/PVP	Before Testing (g)	After the Test (g)	Mass Loss (g)	Mass-Loss Rate
2/2	0.093	0.069	0.027	29.03%
3/3	0.170	0.133	0.037	21.76%
4/4	0.266	0.217	0.049	18.42%

**Table 3 nanomaterials-10-01723-t003:** The tensile strength of the nanofilms.

RL/PVP(g)	Wide (mm)	Thickness (mm)	Peak Load (N)	Tensile Strength (MPa)
0/3	10	1.63 × 10^−2^	0.251	1.540
2/2	10	7.339 × 10^−3^	0.686	9.347
3/3	10	2.002 × 10^−2^	0.889	4.441
4/4	10	3.475 × 10^−2^	1.125	3.2337

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
