# Peer review of "A Study on the Improvement of Using Raw Lacquer and Electrospinning on Properties of PVP Nanofilms"

_nanomaterials, 2020, doi:10.3390/nano10091723_

Round 1

Reviewer 1 Report

In the present manuscript entitled “A study on the Improvement of Using Raw Lacquer and Electrospinning on Properties of PVP Nanofilms” authors has prepared raw lacquer/polyvinyl pyrrolidone (RL/PVP) nanofilms using electrospinning and studies the effect of various factors such as RL/PVP ratio, voltage, flow rate, needle type, and syringe-collector distance on morphology of nanofilms. The manuscript is very poorly written and highly unorganized. Figures resolutions must be improved effectively. I do not recommend this manuscript for publication in Nanomaterials. Few major comments/suggestions are mentioned below.

  1. Author must do proof-reading of this manuscript. Results are highly unconnected and unorganized particularly at the end.
  2. Line-51: What does DOI stands for?
  3. Line- 52: “modifified”- spelling error.
  4. Line 116/117: “FIR, FI-IR”- It must be FT-IR.
  5. Line 143/144/145---0.5ml/h, 10cm (There should be space between numerical value and unit. Please do this for the whole manuscript).
  6. Figure 2 caption: There is no need to write mass unit along with ratio.
  7. Figure 3: Kindly use all SEM images with same scale or magnification for better comparison.
  8. Kindly use same ratio notation throughout the manuscript (For e.g. somewhere it is RL/PVP while at some places it is RL:PVP).
  9. Line 190-Figure 5 (it must be Figure 4).
  10. Figure 4: y-axis caption (units must be in either degree or radian not in °C).
  11. Line-197/199—there is only Figure 4 in the manuscript (not 4(a) and 4(b)).
  12. Section 3.5: Figure numbering and arrangement is poorly presented throughout the manuscript (e.g. Author discuss Fig. 7 first and the switch to Fig. 10 and then start discussing Fig. 9).

Overall, the manuscript needs revamping comprehensively in all parts whether it is writing, Figures, explanations, or presentation.

Author Response

Dear reviewer

    Thank you for your letter dated August 5. We thank the reviewer for the time and effort that they have put into reviewing the previous version of the manuscript. You suggestions have enabled us to improve our work. Based on the instructions provided in your letter, we have uploaded a copy of the original manuscript with all the changes highlighted by using the track changes mode in MS Word.

    Appended to this letter is our point-by-point response to the comments raised by the reviewer. The comments are reproduced and our responses are given directly afterward in a different color (red). We hope that the revised manuscript is accepted for publication in “Nanomaterials”.

Sincerely, 

Qi Lin

Corresponding author:
   Name: Qi Lin

  1. Author must do proof-reading of this manuscript. Results arehighly unconnected and unorganized particularly at the end.

    The typesetting of the article were modified. We also added the cross-section SEM pictures of nanofilms to demonstrate the inner morphology of the samples, that is, was dense fibrous struture. In addition, we conducted the tensile testing, as well as performed a water solubility experiment to determine the mass loss rate of PVP and mechanical properties of thin films. Results showed that the addition of raw lacquer improved the mechanical properties of PVP and also playing an effective protective role to alleviate the water soluble phenomenon of PVP fibers. In conclusion, we investigated the RL/PVP nanofilms forming process and confirmed that it’s possibility to prepare RL/PVP composite fibers with more excellent properties than traditional PVP fibers.

  1. Line-51: What does DOI stands for?

    We have added an explanation to the word "DOI" in the article.

  1. Line- 52: “modifified”- spelling error.

    The word “ modifified ” has been corrected as “ modified ”.

  1. Line 116/117: “FIR, FI-IR”- It must be FT-IR.

    We modified the word according to the comment.

  1. Line 143/144/145---0.5ml/h, 10cm (There should be space between numerical value and unit. Please do this for the whole manuscript).

    We have revised the format for the whole manuscript according to your comments.

  1. Figure 2 caption: There is no need to write mass unit along with ratio.

   We've removed the units in Figure 2 caption.

  1. Figure 3: Kindly use all SEM images with same scale or magnification for better comparison.

    We have provided the clearer images and kept the magnification consistent.

  1. Kindly use same ratio notation throughout the manuscript (For e.g. somewhere it is RL/PVP while at some places it is RL:PVP).

    We have used the same symbol “ RL/PVP ” throughout the manuscript.

  1. Line 190-Figure 5 (it must be Figure 4).

    We have revised the error according to the comments.

  1. Figure 4: y-axis caption (units must be in either degree or radian not in °C).

    We have modified Y-axis caption.

  1. Line-197/199—there is only Figure 4 in the manuscript (not 4(a) and 4(b))

    We have corrected the figure numbering and rearranged the diagram order according to the new experimental content and pictures.

  1. Section 3.5: Figure numbering and arrangement is poorly presented throughout the manuscript (e.g. Author discuss Fig. 7 first and the switch to Fig. 10 and then start discussing Fig. 9).

    We have corrected the figure numbering and rearranged the diagram order according to the new experimental content and pictures.

Reviewer 2 Report

Manuscript Ref: nanomaterials-885848-; Title: A Study on the Improvement of Using Raw Lacquer and Electrospinning on Properties of PVP Nanofilms.

This manuscript describes the electrospinning properties of Raw Lacquer/ PVP blend solution.  There are several points that need to be addressed and revised considerably before acceptance of this paper.  The authors’ were not well-discussed mechanism of the prepared electrospun fibers and their characterizations and stability of fibers. Totally, the obtained results are not sound and interesting. The conclusion obtained is not supported by the results.

Comments

  1. The author(s) needs to revise the Graphical Abstract.
  2. Abstract should be rephrased. Major result output should be incorporated in the abstract section.
  3. General comment: the manuscript has language and typo errors.
  4. Introduction: Introduction section is not well structured. It would be better if the manuscript to be included a brief mention of structural, molecular characterization, compositional, and physicochemical properties of Raw lacquer (RL). Introduction should be rephrased by providing the importance and uniqueness of this paper.
  5. Materials and methods: Provide the details of system, process and ambient parameters of RL/PVP electrospinning.
  6. Figure 4; Please check the label in Y-axis. It is wrongly written.
  7. Fiber size distribution, BET surface area, porosity and mechanical properties of RL/PVP would be provided.
  8. Check figure Legend for Fig. 10.
  9. The mechanism of water resistance capacity of RL/PVP film should be described in detail (chemical or physical or other interaction). Further, the mass loss and time of contact graph should be shown to confirm the water resistance property of RL/PVP film.
  10. FTIR spectrum of Fig. 10 should be checked. X-axis should be % T or absorbance.
  11. Conclusion should be rewritten.

Author Response

Dear reviewer

    Thank you for your letter dated August 5. We thank the reviewer for the time and effort that they have put into reviewing the previous version of the manuscript. You suggestions have enabled us to improve our work. Based on the instructions provided in your letter, we have uploaded a copy of the original manuscript with all the changes highlighted by using the track changes mode in MS Word.

    Appended to this letter is our point-by-point response to the comments raised by the reviewer. The comments are reproduced and our responses are given directly afterward in a different color (red). We hope that the revised manuscript is accepted for publication in “Nanomaterials”.

Sincerely, 

Qi Lin

Corresponding author:
   Name: Qi Lin

  1. The author(s) needs to revise the Graphical Abstract.

    We have modified the Graphical Abstract.

  1. Abstract should be rephrased. Major result output should be incorporated in the abstract section.

    We have included the main results of the experiments in the abstract.

  1. General comment: the manuscript has language and typo errors.

   We have modified the language of the article and reformatted the typesetting.

  1. Introduction: Introduction section is not well structured. It would be better if the manuscript to be included a brief mention of structural, molecular characterization, compositional, and physicochemical properties of Raw lacquer (RL). Introduction should be rephrased by providing the importance and uniqueness of this paper.

    We have added the structure, molecular characteristics, composition and physicochemical properties of raw lacquer (RL) in the article. Now the article has highlighted the importance of this study from the excellent performance of raw lacquer and the novelty of experiments.

  1. Materials and methods: Provide the details of system, process and ambient parameters of RL/PVP electrospinning.

    The system, process and environmental parameters of RL/PVP electrospinning have been explained in more detail.

  1. Figure 4; Please check the label in Y-axis. It is wrongly written.

    We have modified Y-axis caption.

  1. Fiber size distribution, BET surface area, porosity and mechanical properties of RL/PVP would be provided.

    We have supplemented the mechanical properties, fiber size distribution in the manuscript. In addition, we conducted the cross-section SEM image of the samples, as well as performed a water solubility experiment to determine the mass loss rate of PVP. However, the porosity of films with RL/PVP=2/2, RL/PVP=3/3, and RL/PVP=4/4 were larger than 100 nm. Therefore, it is difficult to conduct a test on samples with such large porosity. Accordingly, only a brief explanation, but not specific values, were given.

  1. Check figure Legend for Fig. 10.

    The Y-axis caption was wrong and we have modified it.

  1. The mechanism of water resistance capacity of RL/PVP film should be described in detail (chemical or physical or other interaction). Further, the mass loss and time of contact graph should be shown to confirm the water resistance property of RL/PVP film.

    We have explained the water-resistance mechanism of RL/PVP films, and accordingly carried out water solubility experiment to determine the mass loss. The experimental results are shown in the manuscript.

  1. FTIR spectrum of Fig. 10 should be checked. X-axis should be % T or absorbance.

    The Y-axis caption was wrong and we have modified it.

  1. Conclusion should be rewritten.

    We have added the mechanical properties, the cross-section SEM image of the samples in the manuscript. In addition, we conducted the tensile testing, as well as performed a water solubility experiment which can more fully illustrate the experimental conclusions. We have revised and supplemented the conclusion of the article.

Reviewer 3 Report

The results presented in the paper are interesting. However, I have some questions and suggestions for improving the paper.

1) Line 25: the word “exhibit” is double repeated in one and the same sentence. Please, correct it!

2) Line 52: please, decipher the word “DOI”!

3) Line 72: in the word “naofibrous” the letter N is missed.

4) Line 116: “The FIR…” – the letter T is missed.

5) Line119: “…an acceleration voltage of 25 kV” – in the SEM images 5 kV is indicated. Please, correct it!

6) Lines 115-122: Were the samples sputtered by any metal before SEM investigations?

7) Line145: “#23” – what does it mean?

8) Figure 2: a)“…(6000x)…“ – it is useless information. Please, remove it!

9) Figure 3: Please, put well seen scale bars in the SEM images!

10) Please change the places of the figures in the manuscript so that they are described sequentially!

11) Lines 197-199: in the Figure 4 there are no “a” and “b” parts. It seems to be so that the Fig.4 and Fig.5 are confused.

12) Figure 9: Please, put the SEM images with the same magnification!

Conclusion: the manuscript needs minor revision in order to be accepted to publishing in the journal.

Author Response

Dear reviewer

    Thank you for your letter dated August 5. We thank the reviewer for the time and effort that they have put into reviewing the previous version of the manuscript. You suggestions have enabled us to improve our work. Based on the instructions provided in your letter, we have uploaded a copy of the original manuscript with all the changes highlighted by using the track changes mode in MS Word.

    Appended to this letter is our point-by-point response to the comments raised by the reviewer. The comments are reproduced and our responses are given directly afterward in a different color (red). We hope that the revised manuscript is accepted for publication in “Nanomaterials”.

Sincerely,                                                                       

Qi Lin

Corresponding author:
   Name: Qi Lin

  1. Line 25: the word “exhibit” is double repeated in one and the same sentence. Please, correct it!

    We rephrased this sentence according to the comment.

  1. Line 52: please, decipher the word “DOI”!

    We have added an explanation to the word "DOI" in the article.

  1. Line 72: in the word “naofibrous” the letter N is missed.

    The word “ naofibrous ” has been corrected as “ nanofibrous ”.

  1. Line 116: “The FIR…” – the letter T is missed.

    We modified the word according to the comment.

  1. Line119: “…an acceleration voltage of 25 kV” – in the SEM images 5 kV is indicated. Please, correct it!

    We have made modifications according to your comment.

  1. Lines 115-122: Were the samples sputtered by any metal before SEM investigations?

    Since both PVP and RL are non-conductive materials, the platinum splashing was useful for morphology observed by SEM.

  1. Line145: “#23” – what does it mean?

    In this manuscript, we have explained "#23" in detail.

  1. Figure 2: a)“…(6000x)…“ – it is useless information. Please, remove it!

    We have removed the word according to your comment.

  1. Figure 3: Please, put well seen scale bars in the SEM images!

    We have provided the clearer images and kept the magnification consistent.

  1. Please change the places of the figures in the manuscript so that they are described sequentially!

    We have corrected the figure numbering and rearranged the diagram order according to the new experimental content and pictures.

  1. Lines 197-199: in the Figure 4 there are no “a” and “b” parts. It seems to be so that the Fig.4 and Fig.5 are confused.

    We've fixed this error.

  1. Figure 9: Please, put the SEM images with the same magnification!

    We have provided the clearer images and kept the magnification consistent.

Reviewer 4 Report

The manuscript submitted by Wu and coworkers on Lacquer/PVP nanofilms presents a novel study on the use of a natural lacquer in an electrospinning application to produce nanofibrous thin films with good water and acid resistance. This is the first published report that I have seen with successful electrospinning of a lacquer-containing materials. The manuscript is relatively short and the study is somewhat limited but the authors provide coverage of the successful and unsuccessful lacquer/PVP ratios needed for production of the nanofibers. They also demonstrate the improved water and acid resistance of their new nanofibers relative to pure PVP nonofibrous materials. The work is deserving of publication but the presentation needs a reworking to improve the english language and presentation difficulties.

Author Response

Dear reviewer

    Thank you for your letter dated August 5. We thank the reviewer for the time and effort that they have put into reviewing the previous version of the manuscript. You suggestions have enabled us to improve our work. Based on the instructions provided in your letter, we have uploaded a copy of the original manuscript with all the changes highlighted by using the track changes mode in MS Word.

    Appended to this letter is our point-by-point response to the comments raised by the reviewer. The comments are reproduced and our responses are given directly afterward in a different color (red). We hope that the revised manuscript is accepted for publication in “Nanomaterials”.

Sincerely, 

Qi Lin

Corresponding author:
   Name: Qi Lin

The manuscript submitted by Wu and coworkers on Lacquer/PVP nanofilms presents a novel study on the use of a natural lacquer in an electrospinning application to produce nanofibrous thin films with good water and acid resistance. This is the first published report that I have seen with successful electrospinning of a lacquer-containing materials. The manuscript is relatively short and the study is somewhat limited but the authors provide coverage of the successful and unsuccessful lacquer/PVP ratios needed for production of the nanofibers. They also demonstrate the improved water and acid resistance of their new nanofibers relative to pure PVP nonofibrous materials. The work is deserving of publication but the presentation needs a reworking to improve the english language and presentation difficulties.

    We must show grateful for the time and effort that you have put into reviewing the previous version of the manuscript. Your suggestions have enabled us to improve our work. In addition, we have supplemented the content and experiment of this article to make the it more complete. We believe that the conclusion is persuadable now. We are looking forward to seeing our article published on "Nanomaterials".

Round 2

Reviewer 1 Report

The authors revised the manuscript for the typo errors. However, English was not polished carefully.

Author Response

Dear reviewers

    Thank you for your letter dated August 17. We were pleased to know that our work was rated as potentially acceptable for publication in Journal, subject to adequate revision. We thank the reviewers for the time and effort that they have put into reviewing the previous version of the manuscript. your suggestions have enabled us to improve our work.Based on the instructions provided in your letter, we have uploaded a copy of the original manuscript with all the changes highlighted by using the track changes mode in MS Word.   

    We hope that the revised manuscript is accepted for publication in “Nanomaterials”. 

Sincerely,                                         

                                                                                            Qi Lin

Corresponding author:
   Name: Qi Lin

   E-mail: qlin1990@163. com

Reviewer 2 Report

The revised manuscript can be accepted for publication in  Nanomaterials Journal. 

Author Response

Dear reviewers

    Thank you for your letter dated August 17. We were pleased to know that our work was rated as potentially acceptable for publication in Journal. We thank the reviewers for the time and effort that they have put into reviewing the previous version of the manuscript. your suggestions have enabled us to improve our work.Based on the instructions provided in your letter, we have uploaded a copy of the original manuscript with all the changes highlighted by using the track changes mode in MS Word.   

    We hope that the revised manuscript is accepted for publication in “Nanomaterials”.

Sincerely, 

Qi Lin

Corresponding author:
   Name: Qi Lin

   E-mail: qlin1990@163. com
